# Glycaemic Index of Gluten-Free Biscuits with Resistant Starch and Sucrose Replacers: An In Vivo and In Vitro Comparative Study

**DOI:** 10.3390/foods11203253

**Published:** 2022-10-18

**Authors:** Maria Di Cairano, Fideline Laure Tchuenbou-Magaia, Nicola Condelli, Nazarena Cela, Constance Chizoma Ojo, Iza Radecka, Simon Dunmore, Fernanda Galgano

**Affiliations:** 1School of Agricultural, Forestry, Food and Environmental Sciences (SAFE), University of Basilicata, Viale Dell’Ateneo Lucano 10, 85100 Potenza, Italy; 2Division of Chemical Engineering, School of Engineering, Computing and Mathematical Sciences, University of Wolverhampton, Wolverhampton WV1 1LY, UK; 3School of Science, Faculty of Science and Engineering, University of Wolverhampton, Wolverhampton WV1 1LY, UK

**Keywords:** gluten-free, biscuits, glycaemic index, resistant starch, sucrose replacement, in vivo study, in vitro study

## Abstract

The glycaemic index (GI) is used to demonstrate the tendency of foods to increase blood glucose and is thus an important characteristic of newly formulated foods to tackle the rising prevalence of diabetics and associated diseases. The GI of gluten-free biscuits formulated with alternate flours, resistant starch and sucrose replacers was determined using in vivo methods with human subjects. The relationship between in vivo GI values and the predicted glycaemic index (pGI) from the in vitro digestibility-based protocols, generally used by researchers, was established. The in vivo data showed a gradual reduction in GI with increased levels of sucrose substitution by maltitol and inulin with biscuits where sucrose was fully replaced, showing the lowest GI of 33. The correlation between the GI and pGI was food formulation-dependent, even though GI values were lower than the reported pGI. Applying a correction factor to pGI tend to close the gap between the GI and pGI for some formulations but also causes an underestimation of GI for other samples. The results thus suggest that it may not be appropriate to use pGI data to classify food products according to their GI.

## 1. Introduction

The reformulation of gluten-free products in order to improve their nutritional profile and tackle chronic metabolic diseases, such as diabetes and cardiovascular disease, is topical. This include formulations with reduced or zero content of rapidly digestible carbohydrates, decreased readily available glucose-based ingredients and higher level of fibre to reduce the glycaemic index (GI) [1,2,3,4]. Indeed, it has been reported that gluten-free foods have a higher GI when compared to their gluten-containing counterparts [1,2,3]. Correlations were found between celiac disease and type I diabetes, which also justify the importance of providing gluten-free products with a lower GI among the population with celiac disease [5,6,7,8]. Moreover, the increased incident rate of celiac disease in many Western countries [9] also highlights the pressing need to provide gluten-free products with a lower GI. A GI classification system categorizes foods as having a low (55), medium (55 to 70), or high GI (>70) [10]. Although there are a growing number of studies related to the reduction of GI and their evaluation, with more products being offered on the market there is still an on-going debate about the usefulness of the GI classification and the real effect of a low GI on human diet [11,12,13]. Nevertheless, it is well established that a high GI diet might increase the risk of diabetes, obesity and certain types of cancer and a low GI diet can be beneficial for human health [14,15,16,17,18,19]. In this context, it is important to determine the GI of any newly developed food product. GI is determined by means of in vivo study involving human subjects according to ISO 26642:2010 [20]. The in vivo determination of GI is quite expensive, time consuming and, even if the protocol is not invasive, ethical concerns are often raised. To avoid these drawbacks, in vitro protocols are being applied in order to predict the GI [21,22,23,24]. To date, many studies aiming at the evaluation of the characteristics of gluten-free pasta and baked goods in order to understand the effect of new ingredients or processing techniques on the GI have been published [25,26,27].

Biscuits are a convenient food liked by most of the population and they represent a preferred source of carbohydrates for celiacs [28]. They are generally formulated with rice and maize flour and starches [1]. The use of more nutrient-dense flours could help to improve the nutritional quality of gluten-free products, including biscuits. Our research group worked on the formulation of gluten-free biscuits by using unconventional flours. The different phases of the research [29,30,31,32,33] were aimed at the production of appealing low-GI biscuits made with flours that are rarely found in the biscuits available on the Italian market.

Based on the above background, the aim of this work was to investigate the effect of resistant starch and sucrose replacers (inulin and maltitol) on the GI of gluten-free biscuits formulated with buckwheat, sorghum and lentil flours. In addition, the GI in vivo data and the pGI values, previously determined by Di Cairano et al. [33], were compared in order to evaluate the reliability of the data obtained by the in vitro protocol.

## 2. Materials and Method

### 2.1. Samples

The four gluten-free biscuits under study were produced as reported in Di Cairano et al. [33]. The ingredients used in the formulation of the biscuits are presented in Table 1. The control sample with sucrose as a sweetener was formulated with all other ingredients except resistant starch and inulin, whereas other samples had resistant starch and maltitol or inulin as a partial or full replacement of the flours and sucrose, respectively (Table 2).

Di Leo Pietro spa (Matera, Italy) provided the raw materials and the facility for biscuit production. About 100 kg dough were prepared in an industrial spiral mixer, then biscuits were formed through wire cut technology and cooked in a combustion oven tunnel for about 12 min at temperatures ranging between 175 and 235 °C. After cooling biscuits were packed in plastic and paper (81, PapPet) bags. Figure 1 illustrates the production process of the biscuits; all samples and their characteristics are reported in Table 2.

### 2.2. In Vivo Study

A randomized crossover design study was carried out to evaluate in vivo glycaemic responses after the consumption of the experimental biscuits.

All subjects gave their informed consent for inclusion before they participated in the study. The study was conducted in accordance with the Declaration of Helsinki, and the protocol was approved by the Life Sciences Ethics Committee of the University of Wolverhampton (amended LSEC/201920/FTM/81). The amendment was initiated to allow participants to perform the postprandial glycaemic test at home and to mitigate challenges imposed by COVID-19 restrictions. The test was carried out from July to August 2021. It is possible to successfully obtain home glucose profiles by providing participants with a self-monitoring blood glucose device [34]. However, it is important that participants are trained and properly informed.

#### 2.2.1. Subjects

Fifteen healthy subjects were recruited as volunteers from the student and staff population of the University of Wolverhampton (UK) by advertisement. The criteria used to select volunteers were based on the literature [35,36]. These included a body mass index between 18.5–24.9 kg/m^2^, fasting glucose levels within the normal range (<6 mmol/L), non-smoking, no food allergies or intolerance and no use of supplements/medications, such as birth control pills, anti-asthmatic, or diuretics, that might interfere with glucose and lipid metabolism. Furthermore, they should not, within the past 6 months or at the time of the test, have been following a vigorous exercise regimen or weight loss program; have no acute illness, any type of chronic metabolic disease and have had no major medical/surgical events; and not be pregnant or lactating or planning pregnancy within the next 3 months. Four subjects dropped out before the beginning of the study or during the test. Six males and five females took part to the study and their mean average age was 27 ± 5 (range age, 20–45) years and their BMI 20.62 ± 1.91 (18.5–21.51) kg/m^2^. However, the remaining 11 participants provide a reasonable degree of power and precision for investigating the GI of the newly developed formulations, as the generally acceptable size is a minimum of 10 subjects.

#### 2.2.2. Study Design

The study was designed in a way that each subject performed seven tests in a random order. Each subject was asked to visit the University twice, once at the outset for the straining and screening session and the collection of the experimental pack, and once at end of the test for the submission of the clinical bin. The experimental pack consisted of four types of biscuits coded with a set of numbers, three sachets of 55 g of D-(+)- glucose monohydrate powder (Sigma-Aldrich, St. Louis, MO, USA), seven bottles of natural mineral water (Volvic^®^, Danone Waters Ltd., London, UK) a glucose meter kit containing a glucose meter (BGM-60, Romed^®^, Wilnis, The Netherlands), test strips (BGTS-200, Romed^®^), and blood lancets (BL-100TB; Romed^®^), clinical bin (Sharps Bin 1 Litre Mauser UK T/A Daniels Healthcare, Littleborough, UK), sterile cotton wool (Johson & Johnson, New Brunswick, NJ, USA), hand sanitiser (Dettol, Reckitt Benckiser Group PLC, Slough, UK), and participants’ information with a risk assessment sheet, self-administered 24-h dietary recall, detailed instructions regarding how to perform the test, and a results recording sheet. The 55 g of D-(+)- glucose monohydrate (Sigma-Aldrich, USA) was used as a reference, and the portion size of the experimental biscuits was standardized to yield 50 g of available carbohydrate content according to FAO/WHO recommendations [37]. Each subject consumed either the reference beverage (a 50 g glucose in 250 mL of natural mineral water (Volvic, UK)) or one of the test biscuits, as seen in Figure 2.

Test 1, Test 4, and Test 7 were reference solutions. Tests 2, 3, 5, and 6 were the test foods. The presentation order of the test food was randomized for each subject. Each product was identified by a three-digit random code. They self-monitored their blood glucose for two hours using the provided meter kit (Romed^®^, The Netherlands). There was a minimum two-day interval between tests. The blood measurement was taken before starting the test and again after 15, 30, 60, 90, and 120 min. An amount of 250 mL of mineral water was consumed during the test, at a comfortable pace within 15 min from the beginning of the test.

Participants were advised to eat a regular evening meal followed by a 12 ± 1 h overnight fast. They were required not to consume alcohol before or during the study day and refrain from intense physical activity.

#### 2.2.3. Glycaemic Index and Glycaemic Load

The measured glycaemic values were used to build the curve of glycaemic response for each subject and for every sample, including the glucose standard. Then, the incremental area under the blood response curve (iAUC) was calculated using the trapezoidal rule, ignoring the area beneath the curve [20,37]. For each subject, the GI was calculated as follows:$$\mathrm{GI}=\left(\frac{{\mathrm{iAUC}}_{\mathrm{test}\;\mathrm{food}}}{{\mathrm{iAUC}}_{\mathrm{glucose}\;\mathrm{standard}}}\right)\times 100$$

The GI of each test food was taken as the mean of the whole group. The glycaemic load (GL) of a serving was calculated according to the equation [38]:$$\mathrm{GL}=({\mathrm{GI}}_{\mathrm{test}\;\mathrm{food}}\times \mathrm{available}\;\mathrm{carbohydrate}\;\mathrm{in}\;a\;\mathrm{serving}\;\left[g\right])/100$$

The total available carbohydrate data were previously determined [33] and calculated using the databased method.

The serving size of the biscuits was determined based on Council for Research in Agriculture and Agricultural Economy Analysis (CREA) Italian dietary guidelines [39] with a standard food portion size of 20 g per serving.

The obtained in vivo GI values were compared to the previously obtained data from in vitro digestion [33]. The static in vitro digestion protocol was carried out according to Minekus et al. [23] with slight modifications [33] to evaluate the glucose release during digestion and the predicted glycaemic index (pGI).

### 2.3. Data Analysis

Statistical analysis was performed in Excel 2013 (Microsoft Office, Microsoft Corporation, Redmond, WA, USA) using the XLSTAT Premium Version (2019.4.2, Addinsoft, Paris, France). One-way ANOVA was used to compare samples means. ANOVA was followed by Tukey’s HSD test at a 95% confidence interval. Pearson’s correlation test at a 95% confidence interval was used to explore correlations between parameters.

Based on the number of subjects taking part in the study, the obtained data refers to 11 replicates for the biscuit samples and 11 × 3 replicates for the glucose reference standard. Data are thus represented as average ± standard error of mean (SEM).

## 3. Results and Discussion

### Glycaemic Index and Glycaemic Load

The release profile of glucose during the 120 min of observation was different for all the investigated samples (Figure 3). As generally expected, all the samples including glucose reference solution, showed the highest release of glucose at 30 min. The glucose released in the first 30 min represents the rapidly available glucose which has a significant impact on the GI of the product [22,40]. After the peak in the first 30 min, the blood glucose level decreased more or less rapidly, depending on the type of samples. This trend is different from that observed during in vitro digestion [33], where after the first 20 min a plateau or even a small increase in glucose release was recorded [33]. Looking at glucose release graphs reported in the literature, it actually appears that the curve of glucose release reaches a plateau after the peak when samples are digested in vitro, whereas blood glucose measured in vivo decreases after the peak, generally reached at 30 min [41,42,43,44]. These results are an indication of different dynamics of the glucose release between in vivo study involving human subjects and in vitro digestion. Indeed, it is too intricate to exactly reproduce in vivo digestion by means of in vitro methods, although new techniques, such as in silico models and instrumentations (SHIME^®^, Prodigest, Gent, Belgium; TIM, TNO, The Hague, The Netherlands), are being applied and becoming increasing popular in order to conduct in vitro digestion that better mimics in vivo conditions. Unfortunately, those techniques are expensive, and many researchers mostly follow static digestion protocols.

The change in the blood glucose level over time suggested that different GI could be expected for each sample. Indeed, the four samples had a different GI (Figure 4). Control biscuits made with sucrose and without resistant starch were able to be classified as a high GI (77) food product [10], whereas RS-inulin 30 and RS-maltitol 50 made with approximatively 12% resistant starch and 30% inulin replacing sucrose and 50% maltitol replacing sucrose, respectively, could be classified as medium GI foods. In contrast, biscuits formulated with the total replacement of sucrose by maltitol (RS-matitol 100) showed a GI of 33 and can thus be classified as a low GI product. Sucrose replacement with both inulin and maltitol effectively reduced the GI of the biscuits. As expected, the higher the percentage of sucrose replacement, the higher the reduction of the GI.

Regarding inulin and maltitol, they are both considered low GI ingredients [45,46]. Maltitol is included in the list of food additives and no maximum level has been specified for its use. However, foods containing more than 10% of added maltitol, or polyols in general, must bear a warning of possible laxative effects [47,48]. Similarly, no maximum limit has been recommended for inulin use. However, incorporating a high amount of inulin into biscuit formulation made the dough processing very challenging [30]. Inulin is not hydrolyzed in the human digestive tract due to the lack of inulinase; being a non-digestible carbohydrate it can contribute to lowering the blood glucose level [49]. Maltitol is a polyol partially absorbed by the small intestine. Matsuo [50] evaluated glucose responses after the ingestion of 50 g maltitol or of a maltitol:sucrose mixture (50:50). The GI of maltitol was significantly lower than that of sucrose, which was in agreement with other reported data [51]. Other researchers investigated the glycaemic response to glucose and maltitol in three ethnic groups and found that the absorption of these sugars was not affected by ethnicity [52]. To the best of our knowledge, there are no other research papers that have evaluated the effect of sucrose replacement on the GI of gluten-free biscuits by in vivo study with human subjects.

Recently, Atkinson et al. [53], who reviewed and tabulated published and unpublished sources of reliable GI values, suggested that 84% of 135 biscuit references were within low the GI classification, whereas only 12% and 4% were of medium GI and high GI, respectively. Of the reported data, only two samples were indicated as gluten-free biscuits and both were low GI products. These samples were commercial gluten-free “Frollini” (GI = 37) [54] and chocolate-coated gluten-free cookies (GI = 35) (Sydney University‘s Glycemic Index Research Service (Sydney, Australia), unpublished observations). This information seems to contradict the view that gluten-free products have a high GI. For example, Packer et al. [38] reported a high GI for digestive gluten-free biscuits (GI = 83). Gluten-free products generally present a higher GI when compared to their gluten containing counterparts [1,5,55,56]. This is due to their very composition with high content of rapidly digestible starch and lack of gluten network enveloping starch granules. Different studies focusing on the reduction of the GI of gluten-free cereal-based products used in vitro screening methods to calculate pGI or eGI and to evaluate the potential effect that the samples could have on the blood glucose level. Di Cairano et al. [32] evaluated the pGI of gluten-free biscuits made with gluten-free cereals, pseudocereals and legume flours generally little exploited in gluten-free biscuits and compared it with commercially available gluten-free biscuits and a wheat control. They reported a pGI about 23% higher for wheat biscuits and 17% and 30% higher for the two commercially available gluten-free biscuits when compared to experimental formulations. Tartary buckwheat and its malt were used to produce gluten-free cookies, which displayed lower pGI compared to total rice flour control [57]. Sparvoli et al. [58] found a reduction in the pGI of a wheat-maize-based biscuit when replacing the flours with a common bean one. The replacement of maize flour with high amylose maize starch also resulted in the reduction of pGI [59]. Feng et al. [60] reported a significant reduction of the pGI of non-sucrose millet-based gluten-free biscuits when part of the flour and starch were replaced with *Lentinus edoses* powder. All these biscuits [32,57,58,59,60] can be classified as medium-high GI products with pGI values ranging between 57.6 and 100.2. These values appear to contradict the in vivo GI data collected by Atkinson et al. [53] on the GI of conventional and gluten-free biscuits. As mentioned before, data from in vivo measurement of glucose release from gluten-free biscuits are scarcely available. Diverging information are reported about the usefulness of in vitro GI to predict in vivo data. Some authors reported the lack of a strong correlation between in vitro and in vivo data [61]. Others indicated that in vitro hydrolysis index and starch digestion rate are a good predictor of in vivo GI determined in mice and human subjects and that there is a good analogy between the intestinal phase of in vitro digestion protocols and in vivo studies [40,62,63,64]. Nevertheless, in vitro protocols for the quantification of glucose release and subsequent calculation of pGI are considered a reliable method for screening food products and estimating the effect of product reformulations and new processing techniques on their GI.

In this study, a strong correlation was found between GI and pGI values reported in the literature [33] (Table 3). As expected, correlations were also found between free glucose, total sugars and GI. However, results of resistant starch content and rapidly available glucose were not significantly correlated with GI values, displaying a *p*-value < 0.10; this could be due to the small sample size. As previously reported [33], the correlation for sample containing resistant starch was probably affected by the different amounts of sucrose added to the recipe, which were higher in the biscuit with no resistant starch. Previous research on the formulation of conventional and gluten-free biscuits indicated that the use of resistant starch reduces the GI [65,66,67]. The results of the meta-analysis carried out by Afandi et al. [68] confirmed resistant starch as significant factor responsible for the GI of food product. Indeed, it is commonly known that resistant starch contributes to the reduction of blood glucose level after meal. The European Union regulation allows the use of a health claim relating to the reduction in blood glucose levels after a meal when there is 14% of non-digestible starch of total starch present in the product [69]. While acknowledging the fairly small sample size, the correlations found in the study presented here confirm the information and the trends previously reported on food composition and its effect on the GI [40].

Comparing the GI and pGI values [33], it appears that pGI overestimated the GI (Table 4). The tendency of the in vitro glycaemic response to overestimate the in vivo data were previously reported [61]. The percentage increase was not the same for all the samples and was between 9% and 97%. The lower the GI, the higher the overestimation.

On the other hand, some researchers deem that differences found between GI and pGI can be related to the different reference standard employed (glucose vs. white bread) [42]. Based on this consideration, another approach was applied. pGI data [33] were multiplied by 0.7, as previously reported [42,64], in order to correct the use of bread as reference standard instead of glucose. This approach is rarely used in research papers dealing with the calculation of pGI. To the best of our knowledge, it was introduced by Foster-Powell et al. [70] as correction when white bread was used as a reference instead of sucrose. pGI values after the correction was closer to GI values (Table 4). The percentage change was close to 0 for samples with intermediate sucrose replacement levels (30 and 50%) and pGI values of control and total sucrose replacement sample yielded lower when compared to the GI values. From this analysis, it was apparent that the overestimation or the underestimation is not consistent and depends on the type of samples.

Although, in vitro protocols attempt to simulate as closely as possible the in vivo conditions it is difficult to accurately replicate any human digestive conditions. The process of food digestion generally has different phases, including oral digestion, oesophageal transit, gastric digestion, small intestinal digestion, and large intestinal fermentation, and involves complex shear stress and shear rate; gravitational forces; mass transfer, which affects the digestion rate; and the amount of glucose in the blood stream. Moreover, the food matrix controls the rate and extent to which nutrients and bioactive substances become available for absorption, hence controlling the blood’s concentration profile and how well they are utilised by peripheral tissues [71], which explain dependency of GI on the composition or type of the investigated as well as the difference between pGI and GI.

Considering the collected data, it appears that pGI values can be taken into consideration to gather information about the effectiveness of recipe modifications, the use of new ingredients and processing techniques on the GI but not to accurately predict the GI. It does not seem completely appropriate to use pGI data to classify food products according to their GI, even though it is a rapid method, compared to an in vivo study, easily used in many research papers. Nevertheless, pGI is useful in different research and product development phases, as reported by Bohn et al. [72] who suggested that it is wise to continue the optimization of in vitro techniques and use them in research and early stages of product development.

Besides the GI, another important index relating to glycaemic response is the glycaemic load (GL). It was introduced to quantify the overall glycaemic effect of a typical serving of a food product [73,74]. The biscuits of this experiment had a GL per serving comprised between 2.62 and 5.74 (Table 4) with RS-maltitol 100 having the lowest value and the control biscuit with the highest. The four biscuits can all be considered low-GL products, having a GL value < 10 [75]. The values are similar for those published by Henry et al. [75] who reported that GL values comprised between 2.3 and 3.8 for gluten-containing digestive rich tea and oat biscuits. Although GL was introduced to better classify the impact of a specific service of food on the glycaemic response, diverging methods are used to calculate it [53,76] and different portion sizes can be suggested by producers or by nutritional guidelines of different countries. It is therefore difficult to use this index to compare different published data.

## 4. Conclusions

An attempt has been made to compare GI from in vivo study using human subjects and pGI from in vitro digestion protocols for newly formulated gluten-free biscuits containing resistant starch where sucrose was partially or completely replaced by maltitol and inulin. Our study showed that the use of resistant starch and sucrose replacers, as well as inulin and maltitol, effectively reduces the GI of gluten-free biscuits. Strong correlations were found between GI and pGI values, although the significance of this correlation was product formulation-dependent. The values of GI were lower than those of pGI from the literature, which indicates that further studies are needed to improve the in vitro digestion protocol in order to obtain an accurate prediction of the GI. Our findings suggest that the use of flours with a lower GI compared to the ones commonly employed in gluten-free biscuits (maize, rice and starches) is not enough to obtain low GI biscuits.

## Figures and Tables

**Figure 1 foods-11-03253-f001:**
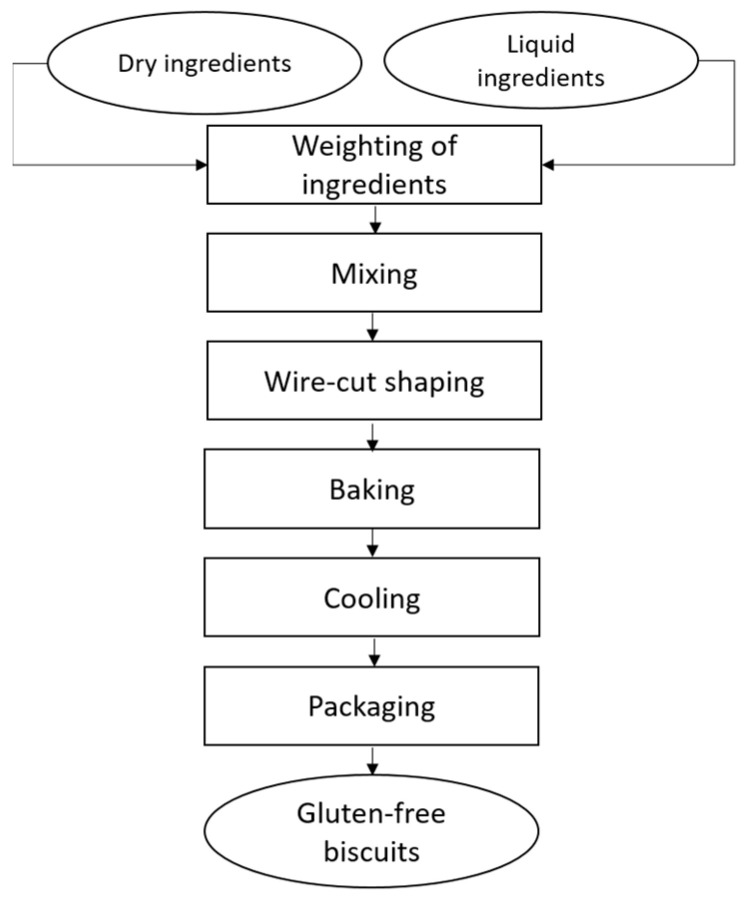
Flow chart of biscuit production process.

**Figure 2 foods-11-03253-f002:**
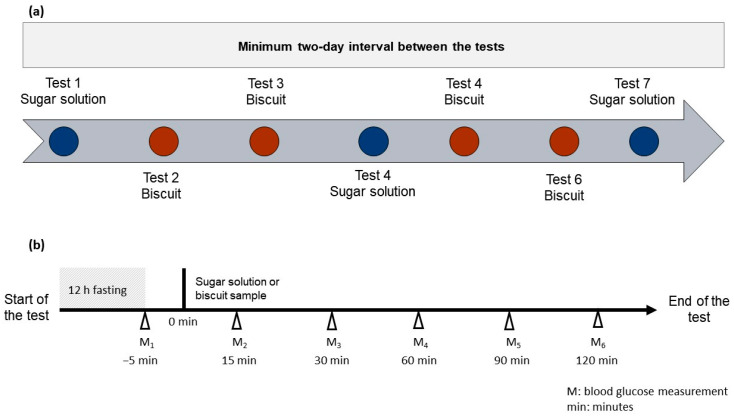
A schematic overview of the seven tests (**a**) and the accompanied visual instruction for performing the test (**b**).

**Figure 3 foods-11-03253-f003:**
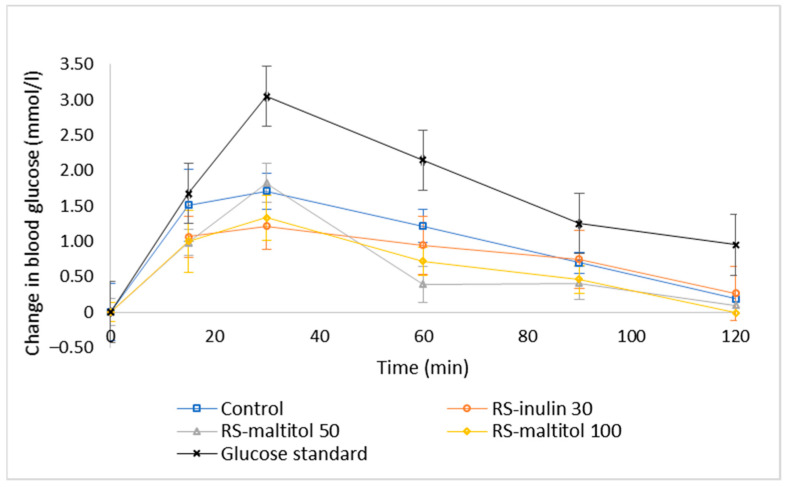
Temporal blood glucose levels for glucose standard and biscuit formulations. Control: biscuits made with buckwheat, sorghum and lentil flours (50:30:20) and 19% sucrose (based on total dough weight); RS-inulin 30, formulated as the Control but with part of the flours replaced by resistant starch (11.50% of total dough weight) and 30% replacement of sucrose with inulin; RS-maltitol 50: formulated as the Control but with part of the flours replaced by resistant starch (12.00% of total dough weight) and 50% of sucrose replaced by maltitol; RS-maltitol 100 as the Control but with but with part of the flours replaced by resistant starch (11.50% of total dough weight) and the total replacement of sucrose with maltitol. Data are reported as mean ± SEM (11 measurements for each biscuit sample and 11 × 3 measurements for glucose standard).

**Figure 4 foods-11-03253-f004:**
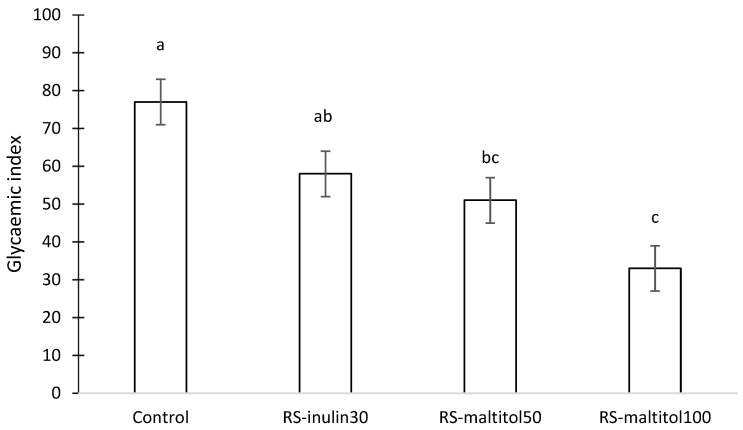
Glycaemic index of gluten-free biscuits made with unconventional flours, resistant starch and sucrose replacers. Control: biscuits made with buckwheat, sorghum and lentil flours (50:30:20) and 19% sucrose (based on total dough weight); RS-inulin 30, formulated as the Control but with part of the flours replaced by resistant starch (11.50% of total dough weight) and 30% replacement of sucrose with inulin; RS-maltitol 50: formulated as the Control but with part of the flours replaced by resistant starch (12.00% of total dough weight) and 50% of sucrose replaced by maltitol; RS-maltitol 100 as the Control but with but with part of the flours replaced by resistant starch (11.50% of total dough weight) and total replacement of sucrose with maltitol. Data are reported as mean value ± SEM. Different letters indicate different means according to Tuckey HSD test (*p* < 0.05).

**Table 1 foods-11-03253-t001:** Ingredients used for the production of gluten-free biscuits samples.

Ingredient	Supplier	g/kg Dough *
Buckwheat flour	Molino Filippini (Teglio, SO, Italy)	279.20
Sorghum flour	Molino Favero (Padova, PD, Italy)	169.50
Lentil flour	Terre di Altamura (Altamura, BA, Italy)	109.70
Resistant starch-HI-MAIZE^®^ 260	Ingredion (Westchester, IL, USA)	-
Sucrose	Suicrà (Pigna Spaccata, NA, Italy)	189.40
Maltitol-Maltite 100	Tereos (Moussy-le-Vieux, France)	-
Inulin–FibrulineTM Instant	Cosucra groupe Warcoing s.a., (Warcoing, Belgium)	-
Eggs	Parmovo (Colorno, PR, Italy)	134.60
High oleic sunflower oil	Tampieri Financial Group (Faenza, RA, Italy)	89.70
Water		19.90
Ammonium bicarbonate	Esseco (Trecate, NO, Italy)	4.00
Sodium hydrogen carbonate	Esseco (Trecate, NO, Italy)	3.00
Salt		1.00

* reported amounts refers to those used in the formulation of control biscuit.

**Table 2 foods-11-03253-t002:** Gluten-free biscuit samples and their main characteristics.

Sample	Characteristics
Control	Buckwheat, sorghum and lentil flour (50:30:20) biscuits with 19% (of total dough weight) sucrose as sweetener and no resistant starch (RS)
RS-inulin 30	As Control but with part of the flours replaced by RS (11.50% of total dough weight) and 30% of sucrose replaced by inulin
RS-maltitol 50	As Control but with part of the flours replaced by RS (12.00% of total dough weight) and 50% of sucrose replaced by maltitol
RS-maltitol 100	As Control but with part of the flours replaced by RS (11.50% of total dough weight) and total replacement of sucrose with maltitol

**Table 3 foods-11-03253-t003:** Pearson’s correlation coefficients between the in vivo glycaemic index and the predicted glycaemic index, resistant starch, rapidly available glucose, free glucose and total sugars.

	Glycaemic Index	
	r	*p*-Value
Predicted Glycaemic Index ^1^	0.992	0.005
Resistant Starch ^1^	−0.984	0.066
Rapidly available glucose ^1^	0.936	0.064
Free glucose ^1^	0.984	0.016
Total sugars ^1^	0.993	0.007

^1^ data used to calculate correlations are from Di Cairano et al. [33]; values in bold indicate significant correlations (*p* < 0.05).

**Table 4 foods-11-03253-t004:** The glycaemic index (GI) predicted glycaemic index (pGI) [33] and glycaemic load (GL) of gluten-free biscuits made with sucrose replacers and resistant starch.

Sample	GI	pGI *	% Change	pGI_G_ **	% Change	GL _(x serving size)_
Control	77 ± 5 ^a^	84 ± 0.4 ^a^	~9	59 **	~(−23)	5.74
RS-inulin 30	58 ± 7 ^ab^	78 ± 0.7 ^b^	~34	55 **	~(−5)	5.17
RS-maltitol 50	52 ± 5 ^bc^	74 ± 0.6 ^c^	~42	52 **	~0	4.05
RS-maltitol 100	33 ± 5 ^c^	65 ± 0.0 ^d^	~97	46 **	~39	2.62

* [33] ** pGI × 0.7 (pGI_G_). Control: biscuits made with buckwheat, sorghum and lentil flours (50:30:20) and 19% sucrose (based on total dough weight); RS-inulin 30, formulated as the Control but with part of the flours replaced by resistant starch (11.50% of total dough weight) and 30% replacement of sucrose with inulin; RS-maltitol 50: formulated as the Control but with part of the flours replaced by resistant starch (12.00% of total dough weight) and 50% of sucrose replaced by maltitol; RS-maltitol 100 as the Control but with but with part of the flours replaced by resistant starch (11.50% of total dough weight) and total replacement of sucrose with maltitol. Data are reported as mean value ± SEM. Different letters indicate different means according to Tuckey’s HSD test (*p* < 0.05).

## Data Availability

The data presented in the manuscript is available upon request.

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
