# Peer review of "Glycaemic Index of Gluten-Free Biscuits with Resistant Starch and Sucrose Replacers: An In Vivo and In Vitro Comparative Study"

_foods, 2022, doi:10.3390/foods11203253_

Round 1
Reviewer 1 Report
The authors investigated the in vivo GIs of gluten-free biscuits with resistant starch and sucrose replacers, and compared them with in vitro predicted GIs. I have some comments as the below.
1. It is better to change the “RS-inulin30, maltitol50, maltitol100” to “RS- inulin 30, maltitol 50, maltitol 100”.
2. It is better to put the explanation of samples (RS-inulin30, maltitol50 and maltitol100) in Fig. 2, Fig. 3 and Table 3 in Section 2.1.
3. Fig.2. Please add the SD in the Figure. In addition, is the blood glucose 0 at 0 min? Please check it.
4. Usually, the *, **, and *** mean p<0.05, 0.01, and 0.001, respectively. The p>0.05 means no significant difference, and has no *.
5. Table 3. Please add the SD and their differences between different samples.
6. It is better to give a concise Conclusion.
Author Response
Glycaemic index of gluten-free biscuits with resistant starch and sucrose replacers: an in vivo and in vitro comparative study
Dear Editor,
We would like to thank the reviewers for their constructive feedback and comments. Each of the questions and pertinent comments of the Reviewers have been duly addressed and incorporated into the revised version of the manuscript using track changes function. For convenience, the responses to each comment are in the following numbered list.
Reponses to reviewer 1
- It is better to change the “RS-inulin30, maltitol50, maltitol100” to “RS- inulin 30, maltitol 50, maltitol 100”.
The change has been done according to the suggestion.
- It is better to put the explanation of samples (RS-inulin30, maltitol50 and maltitol100) in Fig. 2, Fig. 3 and Table 3 in Section 2.1.
The explanation of the samples has been updated in section 2.1 (Table 1). In addition, the description has also been revised to better explain the composition of the samples. The sample description has also been in the legend of Figures and Tables for making them self-explanatory.
- Fig.2. Please add the SD in the Figure. In addition, is the blood glucose 0 at 0 min? Please check it.
Standard error bars have been added to the figure, SEM was added instead of SD to keep consistency with the data already reported in the manuscript. Figure has been colored in order to make it easier to read. Blood glucose at 0 minutes is not 0. It looks like 0 because the area beneath the fasting concentration is ignored when calculating the area under the curve. To avoid misunderstanding, the name of y axis has been changed in “Change in blood glucose” (instead of blood glucose) as also reported by other researchers (i.e. Henry, C. J. K., Lightowler, H. J., Newens, K., Sudha, V., Radhika, G., Sathya, R. M., & Mohan, V. (2008). Glycaemic index of common foods tested in the UK and India. British journal of nutrition, 99(4), 840-845; Pratt, M., Lightowler, H., Henry, C. J., Thabuis, C., Wils, D., & Guérin-Deremaux, L. (2011). No observable differences in glycemic response to maltitol in human subjects from 3 ethnically diverse groups. Nutrition Research, 31(3), 223-228.)
- Usually, the *, **, and *** mean p<0.05, 0.01, and 0.001, respectively. The p>0.05 means no significant difference, and has no *.
The table has been adjusted. All p-values have been indicated in it only those <0.05 have been highlighted as significant.
- Table 3. Please add the SD and their differences between different samples.
SEM has been added instead of SD to keep consistency with the data already reported in the manuscript. No information about the variability of data have been added for pGIG and GL since they have been calculated from the mean pGI and GI value, respectively.
- It is better to give a concise Conclusion.
Thank you for your comment. The conclusion has been reviewed accordingly with information not directly related to the results deleted.
Reviewer 2 Report
The study has been well designed to investigate the effects of different ingredients on GI of gluten free biscuits. Authors have given a thorough comparison of the in-vivo and in-vitro results. In continuation to the previous work of the authors, some methodology is not explained in detail, try to give all the possible details of experimentation.
Authors should incorporate all the corrections and suggestions marked in the pdf file.

Author Response
Glycaemic index of gluten-free biscuits with resistant starch and sucrose replacers: an in vivo and in vitro comparative study.
Dear Editor,
We would like to thank the reviewers for their constructive feedback and comments. Each of the questions and pertinent comments of the Reviewers have been duly addressed and incorporated into the revised version of the manuscript using track changes function. About reviewer 2 observations, the manuscript has been edited according to the comments and amendments reported in the pdf file by the reviewer himself. Below are reported some clarifications for a couple of observation
Reponses to reviewer 2
- what are the permissible levels of these sweetners as per regulations and are these added in view of the permissible limits. Plz mention
There are no limitations in the use of maltitol and inulin in food products. The information has been added in section 3.1 as follows:
“Maltitol is included in the list of food additive and no maximum level has been specified for its use. However, foods containing more than 10% of added maltitol, or polyols in general, must bear a warning of possible laxative effect [47,48]. Similarly no maximum limit has been recommended for inulin use. However, incorporating high amount of inulin into biscuits formulation made the dough processing very challenging [30].”
- Di Cairano, M.; Caruso, M.C.; Galgano, F.; Favati, F.; Ekere, N.; Tchuenbou-Magaia, F. Effect of sucrose replacement and resistant starch addition on textural properties of gluten-free doughs and biscuits. Eur. Food Res. Technol. 2021, 247, 707–718, doi:10.1007/s00217-020-03659-w.
- European Parliament and Council Commission Regulation (EU) No 1129/2011 of 11 November 2011 amending Annex II to Regulation (EC) No 1333/2008 of the European Parliament and of the Council by establishing a Union list of food additives. Off. J. Eur. Union 2011, L295, 1–177, doi:10.3000/19770677.L_2011.295.eng.
- European Parliament and the Concil of the European Union Regulation (EC) No 1333/2008 of the European Parliament ans of the Council of 16 December 2998 on food additives. Off. J. Eur. Union 2008, 16–33.
- Figure 2
Standard error bars have been added to the figure, to keep consistency with other data shown in the paper SEM was added instead of SD. Figure has been colored in order to be easier to read and caption has been updated
- pGI and GI data shown are varying to the extent that the class of GI is also changing e.g., GI of RS maltitol50 is low based on in-vivo measurement and high when done by in-vitro method. plz comment
A comment as been added in section 3.1:
“In vitro and in vivo GI values were different probably due to a different dynamic of digestion as it is difficult to accurately replicate any human digestive conditions. The process of food digestion generally has different phases with oral digestion, oesophageal transit, gastric digestion, small intestinal digestion, and large intestinal fermentation and involves complex shear stress and shear rate, gravitational forces, mass transfer which would affect the digestion rate and the amount of glucose in the blood stream. Moreover, the food matrix controls the rate and extent to which nutrients and bioactive substances become available for absorption, hence controlling the blood's concentration profile and how well they are utilised by peripheral tissues [71] which explain dependency of GI on the composition or type of the investigated samples.”
- Capuano, E.; Janssen, A.E.M. Food Matrix and Macronutrient Digestion. Annu. Rev. Food Sci. Technol. 2021, 12, 193–212, doi:10.1146/annurev-food-032519-051646.
Reviewer 3 Report
Recommendation: Minor Revision
The manuscript Glycaemic index of gluten-free biscuits with resistant starch 2 and sucrose replacers: an in vivo and in vitro comparative study, the methodology were reasonable and technically sound. Here are some issues to be addressed.
Minor concerns:
Comments to the Author:
The manuscript's title is appropriate. Abstract: The Background of the abstract is well written. The main procedure and findings of the study are well expressed. Introduction: A brief survey of existing literature, purpose, importance, and innovation of the research is well mentioned. The tables and graphs are well prepared.
Point 1: I recommend creating a flow chart for gluten-free biscuit production.
Point 2: information of the blood glucose device should be given. The use of different blood sugar devices here reduces the reliability of the study.
Point 3 Please share company information for glucose powder and mineral water.
Point 4: The number of replications of the study should be stated in the statistical part.
Point 5: Figure 2 should be reconsidered. Standard deviations and error bars of the samples are not clear.
Point 6: Line 272, 273, Capitalization errors should be corrected and the sentence should be revised again.
Author Response
Glycaemic index of gluten-free biscuits with resistant starch and sucrose replacers: an in vivo and in vitro comparative study
Reponses to reviewer 3
Point 1: I recommend creating a flow chart for gluten-free biscuit production.
Thank you for your suggestion. A production flow chart has been added in the method section.
Point 2: information of the blood glucose device should be given. The use of different blood sugar devices here reduces the reliability of the study.
Thank you for the pertinent comment. The information about the blood sugar devices used has been added. As this is a home study and in real life people may use the same brand but we rarely use exactly the same glucose meter. We have used the same lot of glucose meter kits bought from the same company and followed the instruction with calibration to minimize any variability due to the device.
Point 3 Please share company information for glucose powder and mineral water.
Glucose powder used was from Sigma Aldrich, the information is reported in line 10 (numbering of original pdf version of the manuscript file) of section 2.2.2. Mineral water was Volvic Natural Mineral Water bought at a UK supermarket.
Point 4: The number of replications of the study should be stated in the statistical part.
The number of replications have been added in “Data analysis” section.
Point 5: Figure 2 should be reconsidered. Standard deviations and error bars of the samples are not clear.
Standard error bars have been added to the figure, SEM was added instead of SD to keep consistency with the data already reported in the manuscript. Figure has been colored in order to make it easier to read. Blood glucose at 0 minutes is not 0, it is reported like because when calculating the area under the curve the area beneath the fasting concentration is ignored. To avoid misunderstanding, the name of y axis has been changed in “Change in blood glucose” (instead of blood glucose) as also reported by other researchers (i.e. Henry, C. J. K., Lightowler, H. J., Newens, K., Sudha, V., Radhika, G., Sathya, R. M., & Mohan, V. (2008). Glycaemic index of common foods tested in the UK and India. British journal of nutrition, 99(4), 840-845; Pratt, M., Lightowler, H., Henry, C. J., Thabuis, C., Wils, D., & Guérin-Deremaux, L. (2011). No observable differences in glycemic response to maltitol in human subjects from 3 ethnically diverse groups. Nutrition Research, 31(3), 223-228.)
Point 6: Line 272, 273, Capitalization errors should be corrected and the sentence should be revised again.
The sentence has been revised and capitalization error removed.
Round 2
Reviewer 1 Report
No